# Unmanned Aerial Vehicle Operating Mode Classification Using Deep Residual Learning Feature Extraction

**Carolyn J. Swinney** [1,2,*] and **John C. Woods** [1]

1 Computer Science and Electronic Engineering Department, University of Essex, Colchester CO4 3SQ, UK; woodjt@essex.ac.uk
2 Air and Space Warfare Centre, Royal Air Force Waddington, Lincoln LN5 9NB, UK
* Correspondence: cjswin@essex.ac.uk

**Abstract:** Unmanned Aerial Vehicles (UAVs) undoubtedly pose many security challenges. We need only look to the December 2018 Gatwick Airport incident for an example of the disruption UAVs can cause. In total, 1000 flights were grounded for 36 h over the Christmas period which was estimated to cost over 50 million pounds. In this paper, we introduce a novel approach which considers UAV detection as an imagery classification problem. We consider signal representations Power Spectral Density (PSD); Spectrogram, Histogram and raw IQ constellation as graphical images presented to a deep Convolution Neural Network (CNN) ResNet50 for feature extraction. Pre-trained on ImageNet, transfer learning is utilised to mitigate the requirement for a large signal dataset. We evaluate performance through machine learning classifier Logistic Regression. Three popular UAVs are classified in different modes; switched on; hovering; flying; flying with video; and no UAV present, creating a total of 10 classes. Our results, validated with 5-fold cross validation and an independent dataset, show PSD representation to produce over 91% accuracy for 10 classifications. Our paper treats UAV detection as an imagery classification problem by presenting signal representations as images to a ResNet50, utilising the benefits of transfer learning and outperforming previous work in the field.

**Keywords:** unmanned aerial vehicles; UAV detection; RF spectrum analysis; machine learning classification; deep learning; convolutional neural network; transfer learning; signal analysis

## 1. Introduction

UAVs are used widely in a manner of different civil applications, from providing wireless coverage for networks [1] to conducting search and rescue missions and even the delivery of goods [2]. In 2020 the global drone market was valued at 22.5 billion USD and is predicted to grow to 42.8 billion USD by 2025 [3]. While UAVs undoubtedly provide many economic and social benefits, they pose equal challenges including the facilitation of crime, aircraft/airport disruption, the delivery of cyber-attacks and physical attacks, for example carrying a software defined radio (SDR) or explosive device. In December 2018 1000 flights were grounded for 36 h at Gatwick Airport due to UAV sightings [4]. This disruption was estimated to cost over 50 million pounds. The facilitation of crime is another area that poses real security challenges, UK prisons have been targeted for drug deliveries using UAVs [5]. Another major concern is the weaponisation of UAVs. Violent non-state actors have been using low cost weaponised UAV technology for years as a way of balancing the power with limited resources in asymmetric warfare [6]. As far back as 2017 former UK Prime Minister David Cameron, warned of the risk of the use of a UAV equipped with an aerosol device for the dispersal of nuclear or chemical material over a European city [7]. In 2020, steps forward have been made to combat the threat in terms of policy, legislation and technology. In 2019 a strategy was published by Government regarding malicious use of UAVs [8] and in January 2020 a police powers bill was introduced for the enforcement of laws surrounding it. Policy and legalities are important but technology is equally looked

upon to provide solutions to UAV misuse [4]. The detection of an unwanted UAV in an airspace is the first step towards its mitigation.

UAV detection has been studied in various forms and can be broken down into the following categories; audio detection, video/image detection, thermal detection, Radio Frequency (RF) detection and RADAR detection. Audio signals have been studied using microphones pointed in different directions that can capture sound up to 30 ft away. Mezei et al. [9] use mathematical correlation as a way of fingerprinting audio signals from different UAVs. This method concentrates on capturing the motor sound which resonates around 40 KHz but this technique struggles in urban areas where the base noise level is high. Recent work in the imagery and video detection field have come up against difficulties in trying to distinguish birds from UAVs, particularly birds such as seagulls that glide and especially at a distance. Shummann et al. in [10] show that video detection methods are making progress with this issue using feature extraction and classification through CNNs. Busset et al. [11] combine the first two approaches with a video and microphone system achieving a detection range of 160–250 m (UAV type dependant). The combination of different detection techniques is thought to warrant further investigation as each technique has different strengths and weaknesses. Andraši et al. [12] show thermal detection to work better on fixed wing UAVs. This is due to quadcopters being built with small electronic motors which are more efficient and therefore harder to detect as they produce little heat. Thermal signatures have not had as much research attention due to high cost, low detection rates and limited distances [13]. RADAR detection methods struggle with the UAV being classed as clutter. This is due to UAVs tending to fly low and be relatively small in size (compared to an aircraft for example). Research in this field looks to improve the detection rate by trying to separate the UAV from the clutter [14]. Lastly, RF detection is seen as one of the most effective ways of detecting UAVs at long distance ranges and has shown detection at 1400 ft [13]. It does warrant its own concerns too, with the UAV often having to be in line of sight with the antenna and some techniques have struggled with interference from other devices which use the same frequency bands such as Wi-Fi and Bluetooth signals. This paper looks to extend current research in the use of RF signals to detect and classify UAVs. We will now discuss current and the associated literature in the field and systems available commercially.

The Drone Detection Grid [15] is a system made by DD Countermeasures, Denver, CO, USA which flags unknown transmitters within a certain frequency range. Systems like these are susceptible to false positives as no classification of the signal is performed, so it could be something other than a UAV flagging detection. Ref. [16] describes a solution based on a network of sensors which use energy detection to identify the UAV and then correlation to classify them. Since the system uses a network of sensors, time difference of arrival can then be used to locate the device. However, due to the number of distributed sensers required these systems can be expensive.

In the recent literature, Nguyen et al. [17] use Wireless Research Platform Board (WARP) version 3 by Mango Communications and the USRP B200 by Ettus Research software defined radio (SDR) platforms for the testing of passive and active RF techniques to detect UAVs. Their work concluded that the UAVs tested were detectable in the RF spectrum when an active signal was successfully reflected, and by passively observing the communication between the UAV and controller. Their work however was limited to only 2 types of UAV. Ezuma et al. [18] evaluate the classification of different UAV controller handsets. Their system is multistage and uses Markov models-based Bayes decision to detect the signal and then machine learning classifiers are evaluated. K-nearest neighbour produced a classification accuracy of 98.13%. Their work extends the field from the detection of a UAV to the classification of a remote control handset type. Huang et al. [19] use a HackRF SDR to detect UAV controllers. They further prove that localisation of the handset is possible using a modified multilateration technique and 3 or more SDRs. Zhao et al. [20] use a USRP to capture the signal amplitude envelope and principal component analysis to feed a set of features into an auxiliary classifier Wasserstein generative adversarial

network. They achieve 95% accuracy with 4 different types of UAV. Training took place in an indoor environment and testing outdoors. Future work looks to use open datasets so these accuracy figures can be compared to other researchers work.

Al-S'ad et al. produce the open DroneRF [21] dataset. A significant contribution to the field, this is the first open dataset for the classification of UAV flight modes. In the associated publication, Al-S'ad et al. [22] use the USRP SDR to capture raw IQ data. I stands for In phase and Q for Quadrature, the I representing the real component of the signal and the Q the imaginary, a complex number. They use a deep neural network (DNN) to classify 3 different UAVs operating in different modes—switched on; hovering; flying with video; and flying without video. Results show classification accuracy to drop significantly when the classes are increased; 99.7% for 2 classes (UAV present or not), 84.5% for 4 classes (UAV type) and 46.8% for 10 classes (UAV type and flight mode). Al-S'ad et al. [22] struggle to classify flight modes accurately using raw data and a DNN, they conclude that an issue may exist with distinguishing between UAVs produced by the same manufacturer. Swinney and Woods [23] use a VGG-16 CNN for feature extraction after using the DroneRF dataset to produce Power Spectral Density and Spectrogram signal representations. They evaluate machine learning classifiers Support Vector Machine, Logistic Regression (LR) and Random Forest and achieve 100% accuracy for 2 class detection, 88.6% for 4 class UAV type classification and 87.3% accuracy for 10 flight mode classifications. Swinney and Woods show treating signal representations as images using a CNN that they are able to distinguish between UAVs produced by the same manufacturer.

Other domains have benefitted from considering a signal as an image. Long et al. [24] use images of wind power curves to detect anomalies in wind turbine data by identifying anomalous data points. They show the method superior to more traditional methods of outlier detection such as k-means. Spectrogram signal representations as images are used for jamming detection in [25] by doing a comparison with a baseline image. This work does not extend to classification of the signal. O'Shea et al. [26] also use spectrograms as images in conjunction with CNN feature extraction to classify wireless signals such as Global System for Mobile Communications (GSM) and Bluetooth. Their work concludes that this method struggles to pick up burst communications as the spectrogram is looking at the frequency changes over a set time period. If the burst didn't happen within the time frame the spectrogram would miss it. In a similar vein UAV signals can hop around the frequency spectrum, potentially making signals harder to detect and classify accurately through spectrogram time domain image representation. Due to the significance of this method in wireless communications, our work in this paper will extend the types of signal representations considered as graphical images presented to a CNN. Viewing a signal in 2D as an image over 1D signals allows a human in the loop to visually identify issues, in some cases providing a contextual understanding. We will also carry on the findings of Swinney and Woods [23], who showed flight mode classification possible with the same manufacturer, by investigating a deeper CNN architecture for classification accuracy. This work does not consider or compare the additional processing power that would be required for the use of a deep CNN compared to the use of 1D data. For example, there would be practical limitations on hardware. However, this is a larger issue with the implementation of DNNs for practical real time and embedded applications which is reviewed in [27].

The approach proposed in this work utilises transfer learning through the use of a pre-trained ResNet50 CNN on ImageNet to extract features from our graphical image datasets of the various signal representations. These features are then classified using machine learning classifier LR. Figure 1 shows the process from the raw signal data in the open DroneRF dataset in block 1 through to the classification of the signal. Block 2 refers to plotting the raw data as Spectrogram, Histogram, Raw IQ constellation and Power Spectral Density (PSD) graphical representations. Block 3 is concerned with extracting features using a ResNet50 CNN which has already been pretrained on the ImageNet database. Lastly block 4 is concerned with evaluating machine learning model Logistic Regression (LR) as the classifier. The rest of the paper is organised as follows; Section 2 introduces the

methodology including an explanation of the signal representations as images, the CNN feature extraction and the machine learning classifier LR. Section 3 presents the results and Section 4 the conclusions from the work.

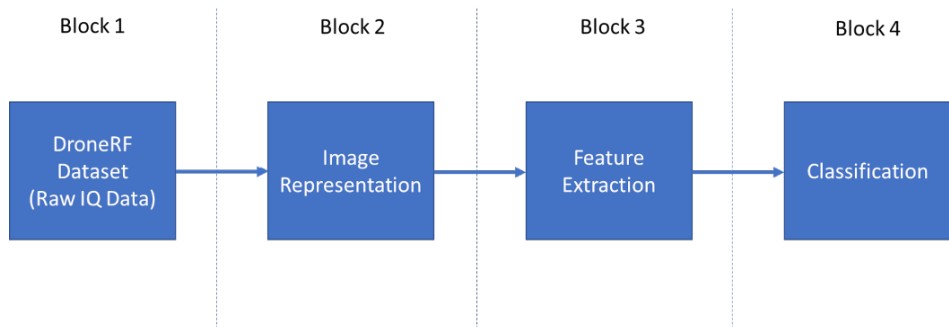

**Figure 1.** Block diagram of the overall model.

## 2. Materials and Methods

### 2.1. Dataset

Table 1 shows UAV classes assessed in these experiments from the open DroneRF dataset produced by Al-Sa'd et al. [21]. Three UAV types are evaluated–Parrot Bebop, Parrot AR (Elite 2.0) and the DJI Phantom 3. Al-Sa'd et al. in [22] pick these UAVs because they are commonly purchased for civilian applications.

**Table 1.** UAV Classes.

| Class | UAV Type | Mode |
| --- | --- | --- |
| 1 | No UAV | N/A |
| 2 | Parrot Bebop | Switched on and connected to controller |
| 3 | Parrot Bebop | Hovering automatically with no controller commands |
| 4 | Parrot Bebop | Flying without video transmission |
| 5 | Parrot Bebop | Flying with video transmission |
| 6 | Parrot AR | Switched on and connected to controller |
| 7 | Parrot AR | Hovering automatically with no controller commands |
| 8 | Parrot AR | Flying without video transmission |
| 9 | AR | Flying with video transmission |
| 10 | DJI Phantom 3 | Switched on and connected to controller |

The Bebop, AR and Phantom 3 are varied when it comes to price, size and overall capability. They increase with weight and size, respectively, and in terms of range the Phantom 3 can operate out to 1000 m, while the Bebop and AR are restricted to 250 and 50 m, respectively. The Phantom 3 and the Bebop utilise the 5 GHz and 2.4 GHz Wi-Fi bands but during these experiments are limited to observing their activity in the 2.4 Ghz band. The Bebop can be set to automatically select a Wi-Fi channel based on the countries legal requirements and channel congestion or you can manually set the channel yourself [28]. Capturing the whole Wi-Fi spectrum ensures that the UAV will be captured even if the device switches channel during operation due to interference.

In Table 1, we can see that for the Bebop and the AR various modes are captured, including switched on and connected to controller; hovering automatically with no input from the controller; flying with video transmission; and flying without video. The DJI Phantom 3 is recorded only in the first mode - switched on and connected to controller. Al-Sa'd et al. [22] is the first work to the authors knowledge which considers different modes of operation when classifying UAVs. The ability to classify the signal would be extremely useful in helping to determine intent. Organisations such as the police could use this information to make an assessment on risk. For example, flying with the video on could indicate an intelligence collection operation due to the real time feedback of imagery.

Intelligence capabilities on UAVs have been directly linked to targeted killing [29]. The DroneRF dataset [21] was recorded using two USRP-2943 SDR made by Ettus Research, Austin, TX, USA. The USRP-2943 is a higher end SDR costing around £6350 each [30]. They operate between 1.2 GHz and 6 GHz frequency range with the ability to capture 40 MHz of instantaneous bandwidth. Al-Sa'd et al. utilise two USRP-2943 simultaneously in order to cover 80 MHz of the Wi-Fi spectrum (excluding channel 1 and 14). For the experiments in this paper 1000 samples were taken for each class and split 80% for training with cross validation and 20% kept entirely separate as a hold-out evaluation dataset.

*2.2. Signal Representation*

2.2.1. Raw IQ Data and Histogram

The raw data captured from an SDR is IQ data. I stands for In phase and Q for Quadrature, the I representing the real component of the signal and Q the imaginary, a complex number. A very basic SDR receiver connected to an antenna is shown in Figure 2 [31].

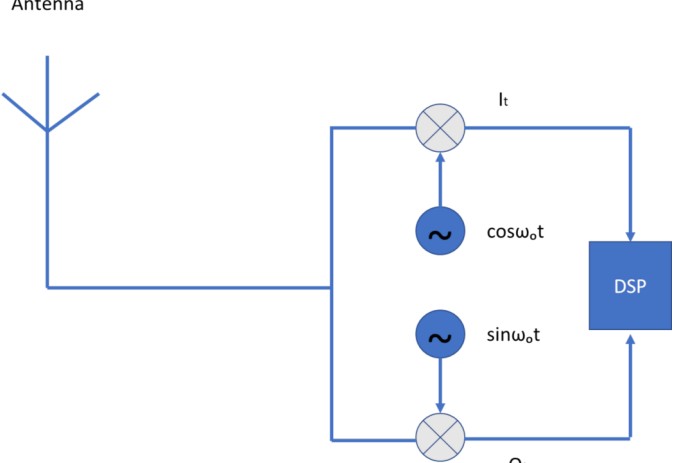

**Figure 2.** SDR Reciever.

In Figure 2, $\omega$ can be equated to $2\pi f$ where f is the frequency from the local oscillator. With respect to time, I and Q components have the same phase relationship as $\sin x$ being 90 degrees different from $\cos x$. Utilising I and Q components allows signals of different frequencies above and below the local oscillation frequency to be separated. There are other advantages as these vectors provide more information for a Fast Fourier Transform (FFT) than a single scaler. Further, it will produce the same result with half the sampling rate so a wider bandwidth can be achieved [31]. I and Q components can be plotted with Matplotlib using a simple scatter plot. A histogram allows us to see a statistical view whereby the occurrences of the real part of the signal are counted over 500 bins and plotted to measure the distribution of the signal.

2.2.2. Power Spectral Density

Power Spectral Density (PSD) calculates the strength of a signal and the distribution of that strength in the frequency domain. This is done using Welch's method, an approach developed by Peter Welch that uses periodogram spectrum estimates and converts the signal from the time to the frequency domain. The Welch method is known for its ability to provide improved estimates when Signal to Noise Ratio is low but there is a tradeoff between the reduction in variables to achieve this and the resolution of the PSD [32]. First the signal in the time domain is partitioned into blocks as shown in Equation (1) ref. [33]. The signal x is broken into m windowed frames and Equation (1) shows the m th

windowed frame from signal x where R is the hop size (the number of samples between each successive FFT window) and K is the number of frames [33].

$$x_m(n) \triangleq w(n)x(n+mR)$$
$$n = 0, 1, \ldots, M-1, \; m = 0, 1, \ldots, K-1 \tag{1}$$

Next the periodogram of the m th block is calculated as show in Equation (2) [33].

$$Px_m, M(\omega_k) \triangleq \frac{1}{M} \left| \sum_{n=0}^{N-1} x_m(n) e^{\frac{-2j\pi nk}{N}} \right|^2 \tag{2}$$

Lastly in Equation (3) we calculate the Welch estimate which is an average of all the periodograms [33].

$$S_x^W(\omega_k) \triangleq \frac{1}{K} \sum_{m=0}^{K-1} P_{x_{m1}M(\omega_k)} \tag{3}$$

Our implementation of PSD uses Python 3 Matplotlib which utilises the Welch method. We use 1024 data points in each segment for the FFT and we include a windowing overlap of 120 points between segments with a Hanning windowing function.

### 2.2.3. Spectrogram

While the PSD looks at the distribution of signal strength in the frequency domain, a spectrogram looks at how the frequencies are changing with time. Spectrograms are used extensively in fields such as speech processing due to their ability to visualise bursts of activities at different frequencies over time. The spectrogram shows the intensity of the Short-Time Fourier Transform (STFT) magnitude over time. It is a sequence of FFTs of windowed data segments which lets us visualise how the frequency content of the signal is changing over time. In Equation (4) we define the STFT [34].

$$
\begin{aligned}
X(\omega, m) &= STFT\,(x(n)) \\
&:= DTFT\,(x(n-m)\omega(n)) \\
&:= \sum_{n=-\infty}^{\infty} x(n-m)\omega(n)e^{-(i\omega n)} \\
&:= \sum_{n=0}^{R-1} x(n-m)\omega(n)e^{-(i\omega n)}
\end{aligned}
\tag{4}
$$

Equation (4) describes a visual representation of the STFT magnitude $|X(\omega, m)|$, the spectrogram. In (4) x(n) represents the signal, $\omega$(n) the windowing function with a length of R and the windowing function determines the block length. As with the PSD, Matplotlib is used to plot the spectrogram with a Hanning windowing function and FFT length 1024.

### 2.3. Image Representation

The frequency range that is covered is from 2.402 GHz–2.482 GHz (Ch 1–Ch 13 Wi-Fi bands with the exception of the first and last 1 MHz). In the figures below 0Hz represents the center of the captured spectrum 2.442 GHz. Figure 3 shows the different signal representations when there is no UAV present. What we are looking at here is the background noise or other signals present in the frequency band at the time of signal capture. Figure 4 shows the Bebop in mode 1–switched on and connected to the controller. It is clear on the PSD that there is some activity in the higher end of the spectrum (2.44–2.48 GHz). As there is no video transmission in this mode, it is likely that this is the command and control signal between the UAV and controller.

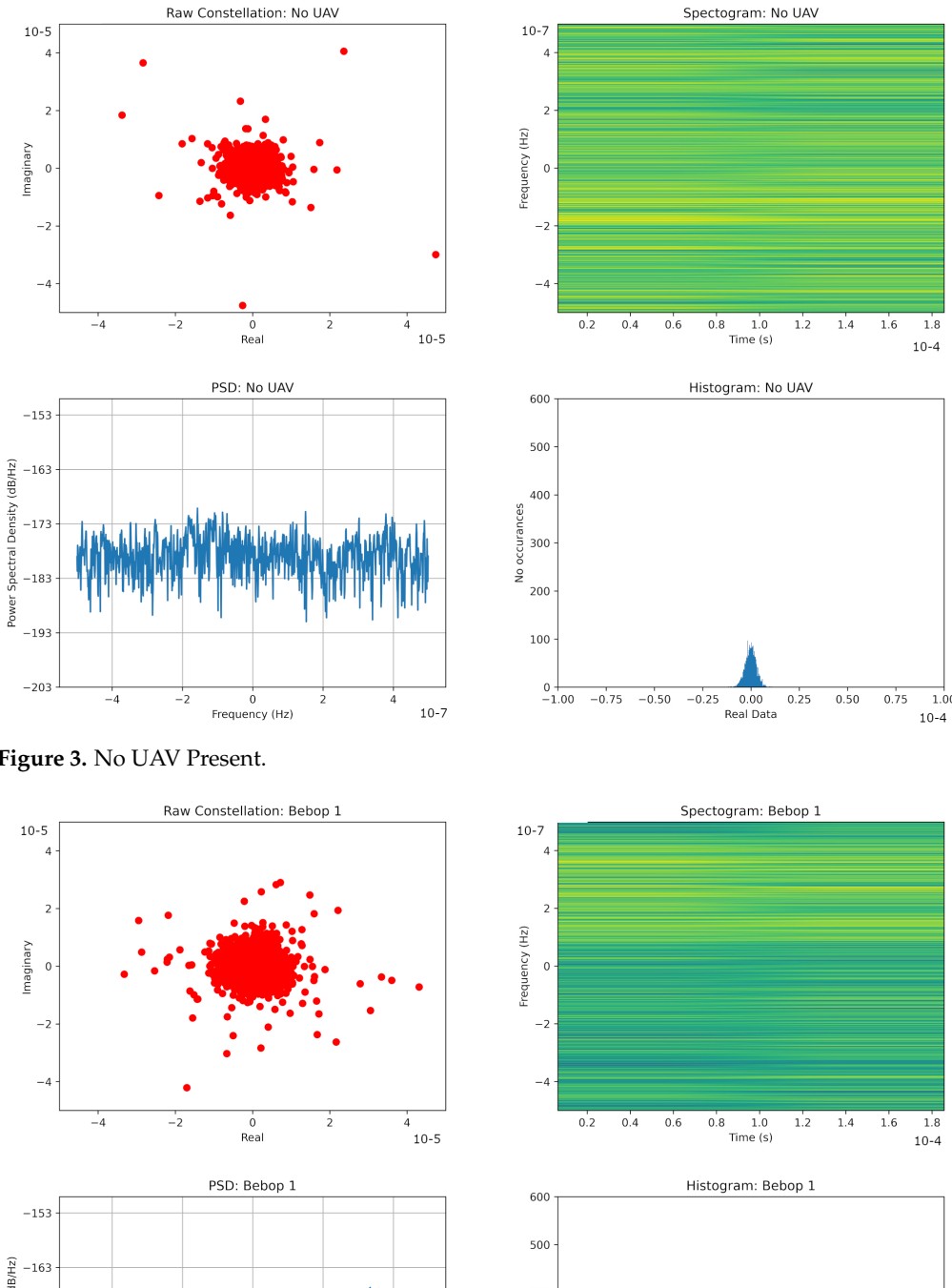

**Figure 3.** No UAV Present.

**Figure 4.** Bebop Mode 1—switched on.

In Figure 5, the Bebop is hovering in automatic mode. What we can see is that there is an even spread of activity across the entire band. In particular if we compare the PSD to the PSD where no UAV is present in Figure 3, we can see there is an increase of around 3 dB across the whole frequency band. If we compare the spectrogram in Figure 5 where

the platform is hovering in an automatic mode (with no active communication with the controller) to Figure 3, we see a decrease in power in the higher end of the spectrum, again indicating this to be a command and control signal.

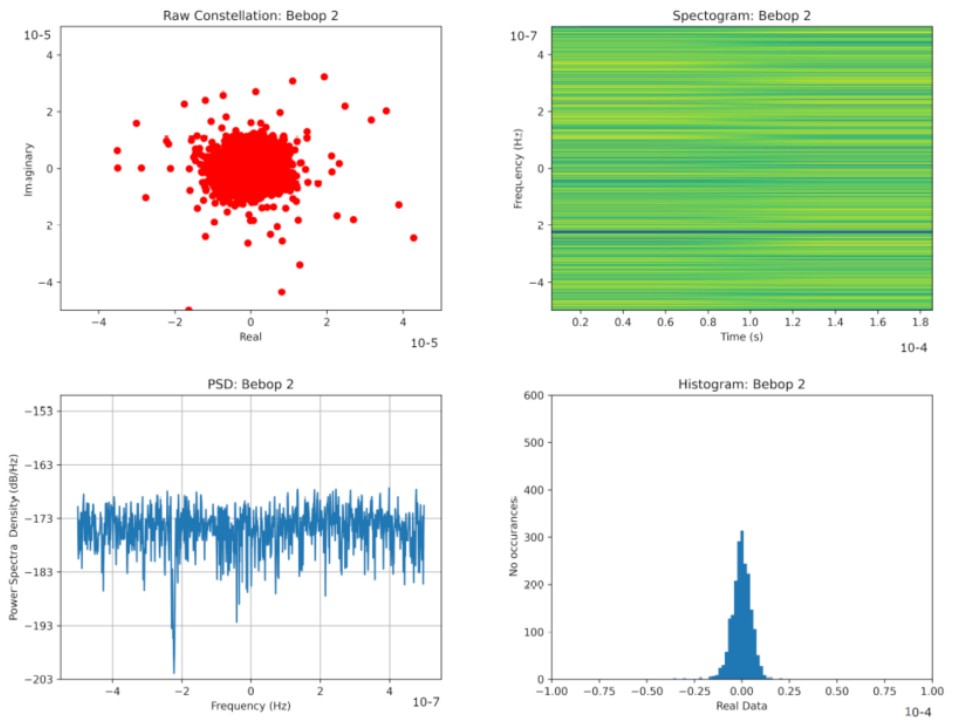

**Figure 5.** Bebop Mode 2—hovering.

Figure 6 shows the Bebop flying without video and Figure 7 with video. The histogram indicates an increase in activity with the video transmitting.

Figure 8 shows the AR switched on and connected to the controller. The spectrogram and PSD show two clear bands of activity in the spectrum, one above and one below the centre frequency. Figure 9 shows the spectrum when the AR is hovering. If we compare this to Figure 5 where the Bebop is also hovering we can see a similar constant spread of activity across the entire spectrum but with a drop at the center frequency.

Figures 10 and 11 show the AR flying without and with video, respectively. We can see in Figure 11 on the PSD that the higher end of the spectrum increases in power by around 3 dB when the video is present. We can also see a clear rise in the histogram representation when the video feed is turned on.

Figure 12 shows the DJI Phantom turned on and connected to the controller. Comparing this to the Bebop in Figure 4 and the AR in Figure 8 we can observe that the Phantom has a more even spread of power across the entire spectrum.

In all of the figures it is hard to see any real pattern in the raw data changes, this may be due to the fact that we looking at so much of the frequency range at once. We need to consider the whole frequency range as we can't be sure if a UAV will hop to a random Wi-Fi channel due to interference during operation. The DroneRF dataset includes 10.25 s of recording with no UAV present and 5.25 s for each UAV flight mode at a sample rate 200 MS/s, producing a dataset larger than 40 GB [22]. Samples are plotted in each of the 4 signal representations using MatPlotlib and saved as images with 300 DPI. We constructed separate datasets of images for raw constellation, spectrogram, PSD and histogram. Each class within each dataset contained 1000 image representations resized to 224 × 224 pixels. The databases were split so there were 8000 images for use with k-fold cross validation and 2000 images with the evaluation set.

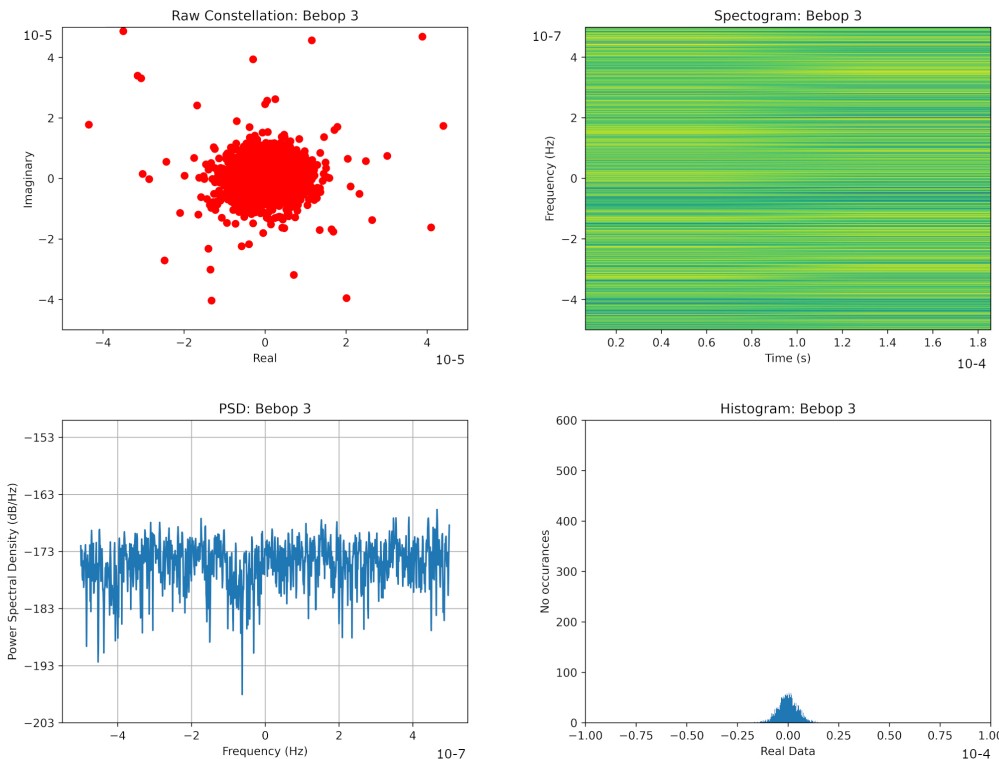

**Figure 6.** Bebop Mode 3—flying without video.

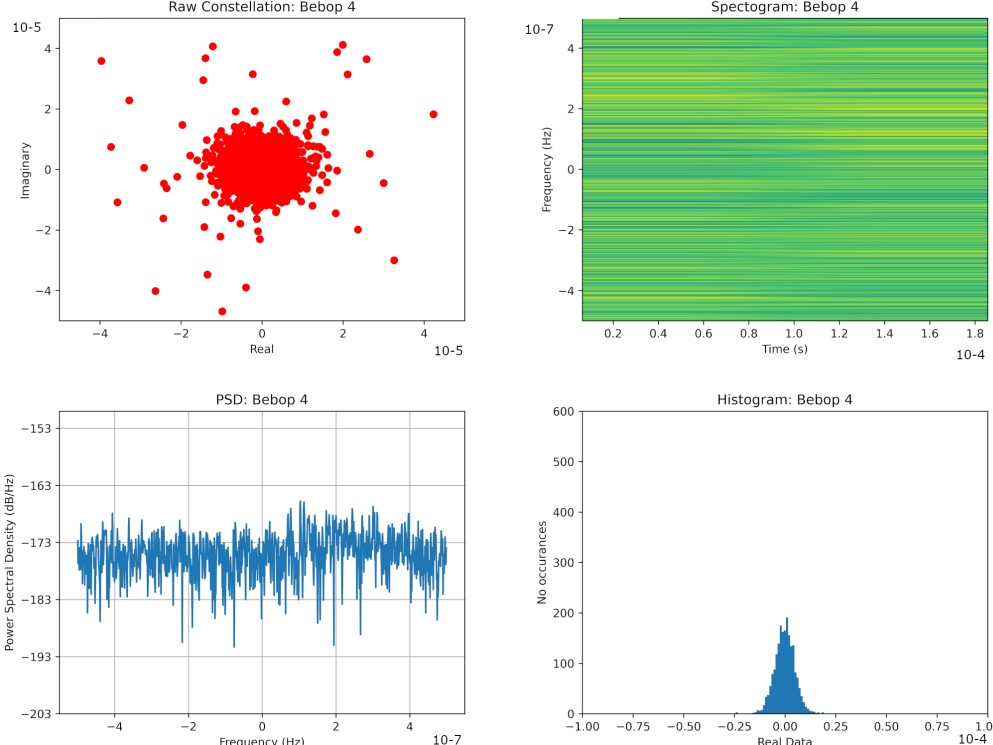

**Figure 7.** Bebop Mode 4—flying with video.

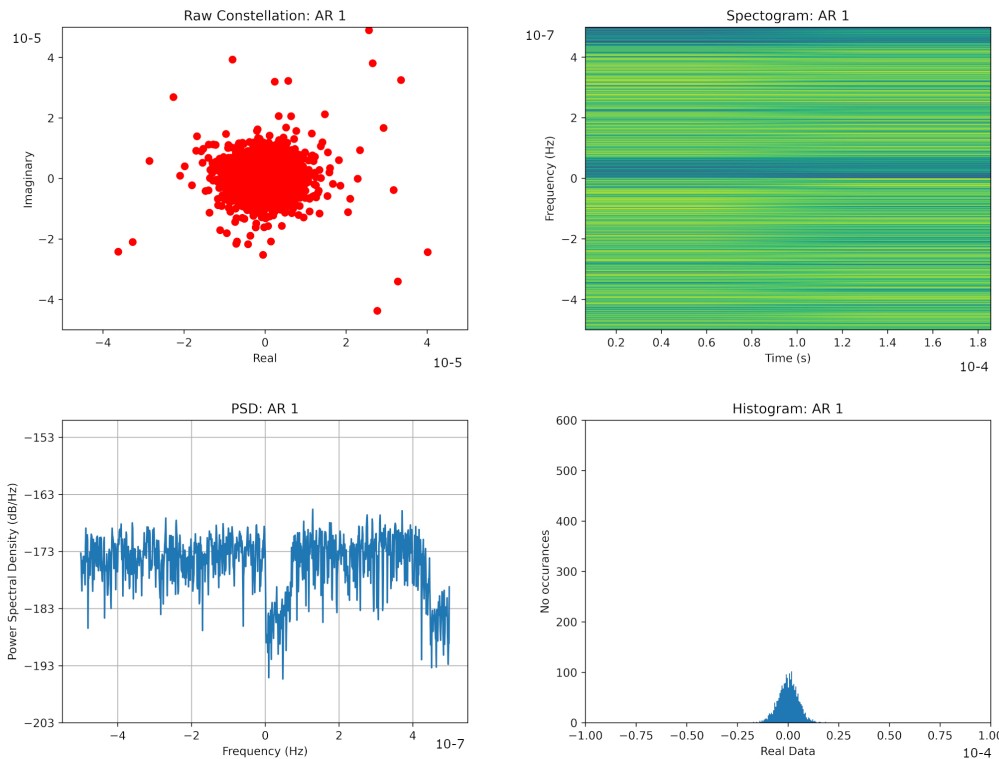

**Figure 8.** AR Mode 1—switched on.

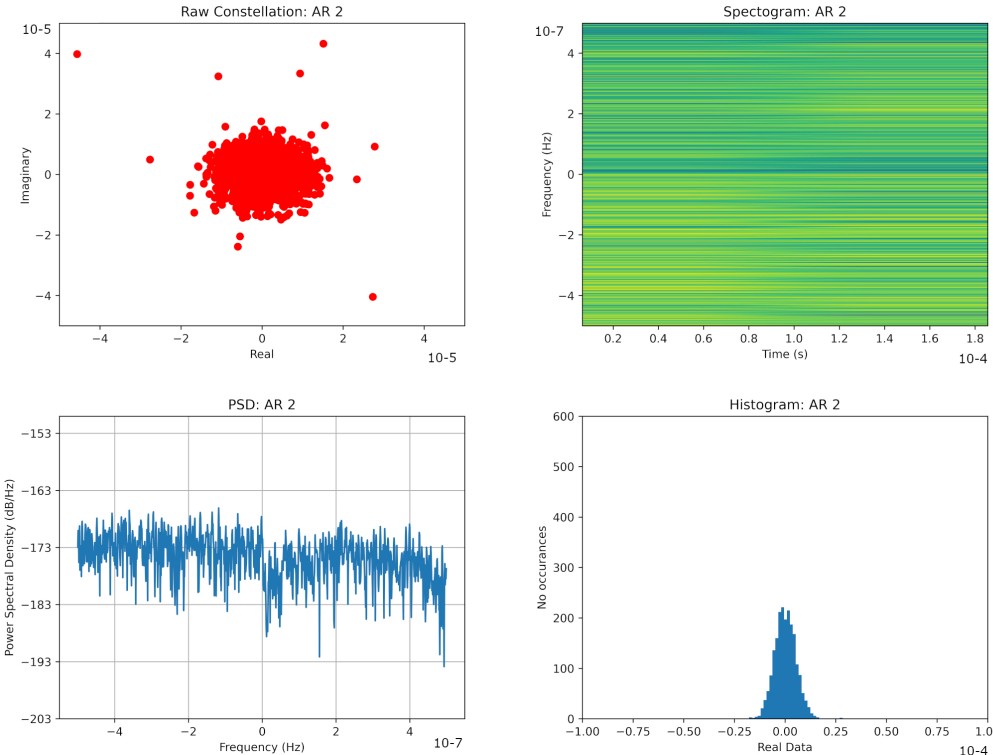

**Figure 9.** AR Mode 2—hovering.

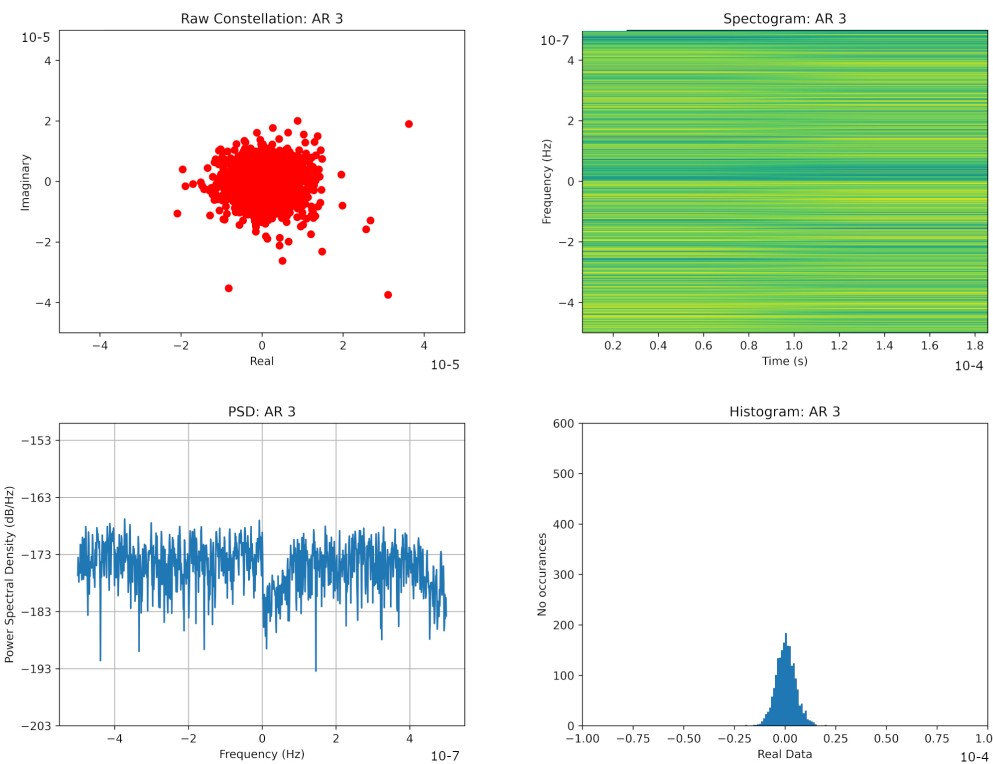

**Figure 10.** AR Mode 3—flying with video.

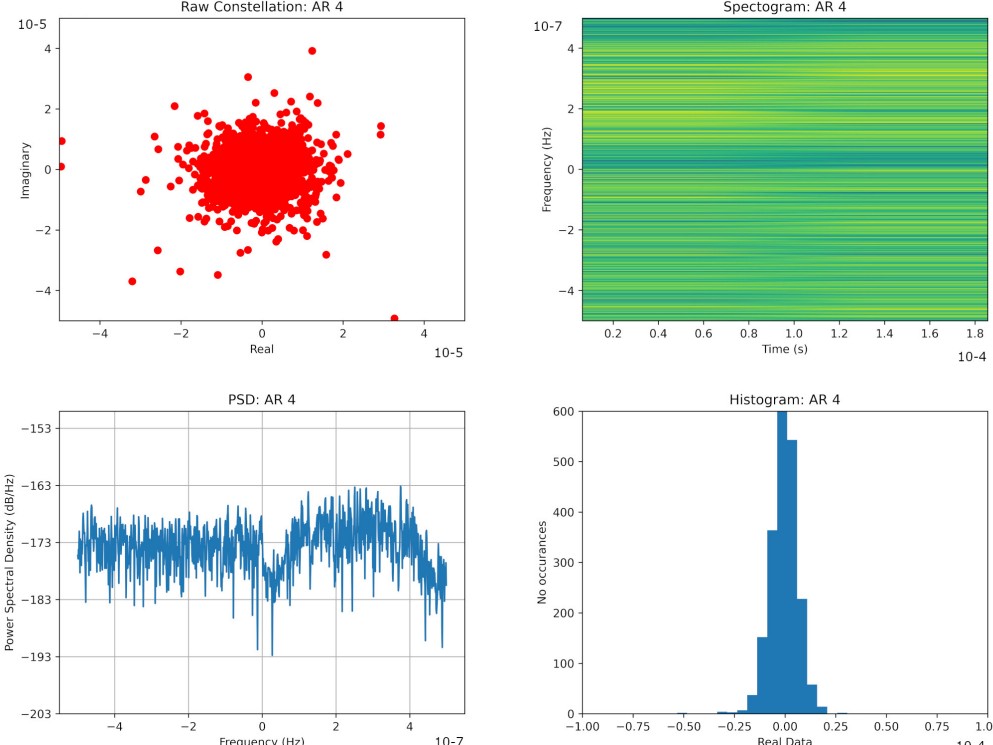

**Figure 11.** AR Mode 4—flying with video.

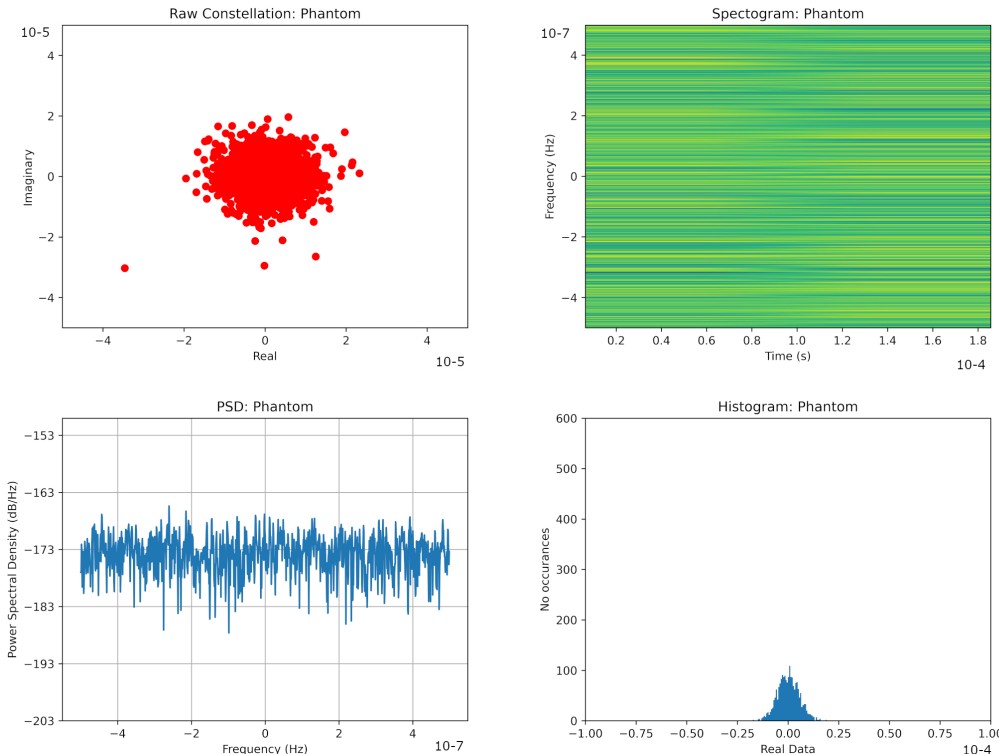

**Figure 12.** AR Mode 3—Phantom Mode 3—switched on.

### 2.4. CNN Feature Extraction

Transfer learning is when a pre-trained network is used for a purpose it was not trained for. It is a popular technique in the medical field for the diagnosis and indication of severity for various medical conditions [35–37]. Residual Networks (ResNet) [38] allows for deep neural networks to be trained using a technique called skip connection which take the output from an earlier layer and combines it with the a later layer. This technique overcame prior difficulties with training very deep neural networks whereby gradients would vanish due to repeated multiplication. ResNet50 has been commonly used for transfer learning research with a large scale image recognition database of over 14 million images called ImageNet [39]. Training the weights in a neural network from scratch can take a very long time and needs a large amount of training data, for example the 14 million images that trained the weights for ImageNet. Transfer Learning allows other domains to benefit from the use of pre-trained weights for a new purpose. In these experiments a CNN ResNet50 trained with ImageNet will be used to extract features from our signal representations, presented to the ResNet50 as 224 × 224 pixel images with 3 channels. ResNet50 is 50 layers deep, consisting of 48 convolution layers, 1 max pooling and 1 average pooling layer. The last layer will have an output shape of 7 × 7 × 2048. This gives us a feature vector of 100,352 values when the shape is flattened.

### 2.5. Machine Learning Classifier Logistic Regression

LR is a machine learning model which has a fixed number of parameters based on the number of features in the input. The output of LR is categorical and uses a sigmoidal curve. The equation for a sigmoid can be seen in Equation (5).

$$h = \frac{e^x}{(1 + e^{-x})} \tag{5}$$

The output of Equation (5) will always be between 0 and 1 so if we define a threshold for example of 0.5, then any values below 0.5 will return 0 and above 1. $x$ represents the

input features and to initialise $\theta$ it is multiplied by a random value $\theta$. When there are multiple features this makes the equation seen in Equation (6).

$$h \;=\; \theta_0 + \theta_1 X_1 + \theta_2 X_2 + \dots \tag{6}$$

The algorithm in Equation (6) updates $\theta$ and eventually will establish a relationship between the features and the output through updating $\theta$. For a situation where we have multiple classes, the sigmoid is generalised and this is called the Softmax function. The Softmax function takes a input vector and then plots it to a probability distribution between 0 and 1. In Equation (7) we describe the softmax function for vector $z$ with $k$ dimensions or classes [40].

$$softmax(z_i) \;=\; \frac{e^{z_i}}{\sum_{j=1}^{k} e^{zj}} \tag{7}$$

LR was implemented in Python 3 through Sklearn. Ridge Regression as the penalty for the loss function and Limited memory Broyden–Fletcher–Goldfarb–Shanno (LGBFS) was used as the solver. Values for regularisation ('C') were optimised using SkLearn GridSearchCV.

### 2.6. Cross Validation

It is important that machine learning models can make predictions on new data, this is called its ability to generalise [41]. Cross validation assesses how a model will generalise and should highlight other problems such as bias or overfitting [42]. Stratified K-Fold cross validation is used in our experiments whereby the training/test data is split into $k - 1$ sets and the sets are used to train the model, except for the last set which is used as test data. Using the stratified version of k-fold in SKlearn allows the same distribution of each class in each subset [43]. The value $k = 5$ was used as it is thought in the statistical community not to be susceptible to either high bias or high variance [44]. Nested cross validation was used with 3 folds in order to optimise hyperparameters. Lastly a hold-out evaluation dataset was kept entirely separate. It was not used to train or test the model so this data provides a further evaluation of the models' performance on unseen data.

### 2.7. Performance Evaluation

There are different metrics we could use to evaluate performance. A confusion matrix helps us to calculate and visualise indictors of performance such as accuracy. Figure 13 shows the components that make up a confusion matrix.

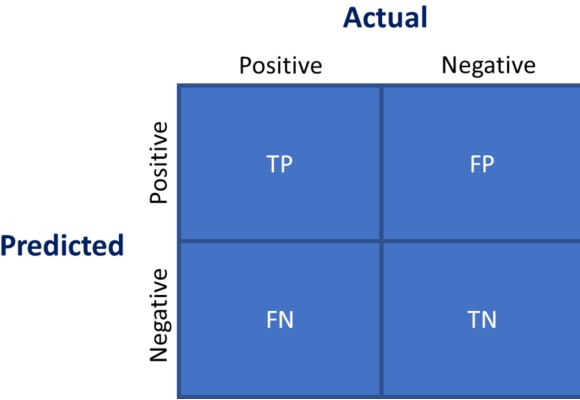

**Figure 13.** Confusion Matrix.

True Positive (TP) tells us that the prediction was correct and it was true to what was predicted. True Negative (TN) is where we have predicted something was incorrect and it was incorrect. False Positive (FP) is where we have predicted something was correct but it was not. False Negative (FN) is where we predicted incorrect when it was correct.

The values help us to define accuracy and F1-score. Accuracy shows us how often the model was right in its predictions, shown in Equation (8).

$$Accuracy \ = \ \frac{TP \ + \ TN}{TP \ + \ TN \ + \ FP \ + \ FN} \tag{8}$$

To calculate F1-Score we need to understand how to calculate both recall and precision. Precision is calculated by dividing TP by TP + FP. This shows how many predicted positives were actually positive. Recall is calculated by dividing TP by TP + FN. It shows the fraction of positives that were correctly predicted. F1-Score is often used as a performance metric as it takes into consideration both recall and precision as seen in Equation (9).

$$F1 \ Score \ = \ \frac{2 \ (Precision \ \times \ Recall)}{Precision \ + \ Recall} \tag{9}$$

## 3. Results

### 3.1. CNN Feature Extraction

To try to understand the features that are being chosen by the CNN we show the output of convolutional layers 0, 20, 40 and 48. We have restricted the depth in the maps to 64 for consistency in the comparison but it should be noted that the depth is greater in deeper layers. PSD was chosen for the feature map visualisation as it produces the highest accuracy in Section 4. Figure 14 shows the input image given to the ResNet50 model for feature visualisation.

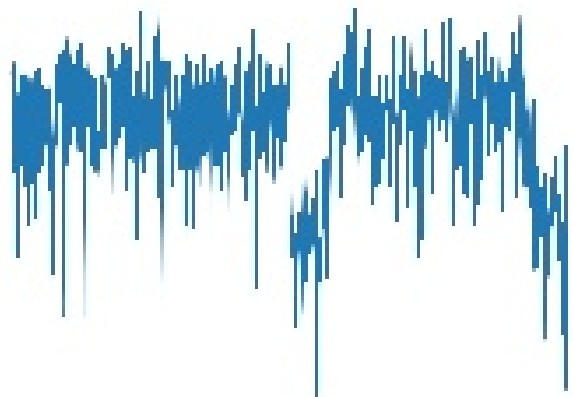

**Figure 14.** Input Image AR Mode 1.

Figure 15 shows the output of the first convolutional layer (layer 0) and we can clearly see the detail of the PSD for both input images AR Mode 1.

As we move to convolutional layer 20, Figure 16 shows that although we can still see the outline and depth of the PSD, we start to lose some detail. This happens because the CNN starts to pick up on generic concepts rather than specific detail.

Figure 17 shows the output of the last convolutional layer 48 and we can see that it is difficult now to determine with the human eye what the features are that the CNN is distinguishing.

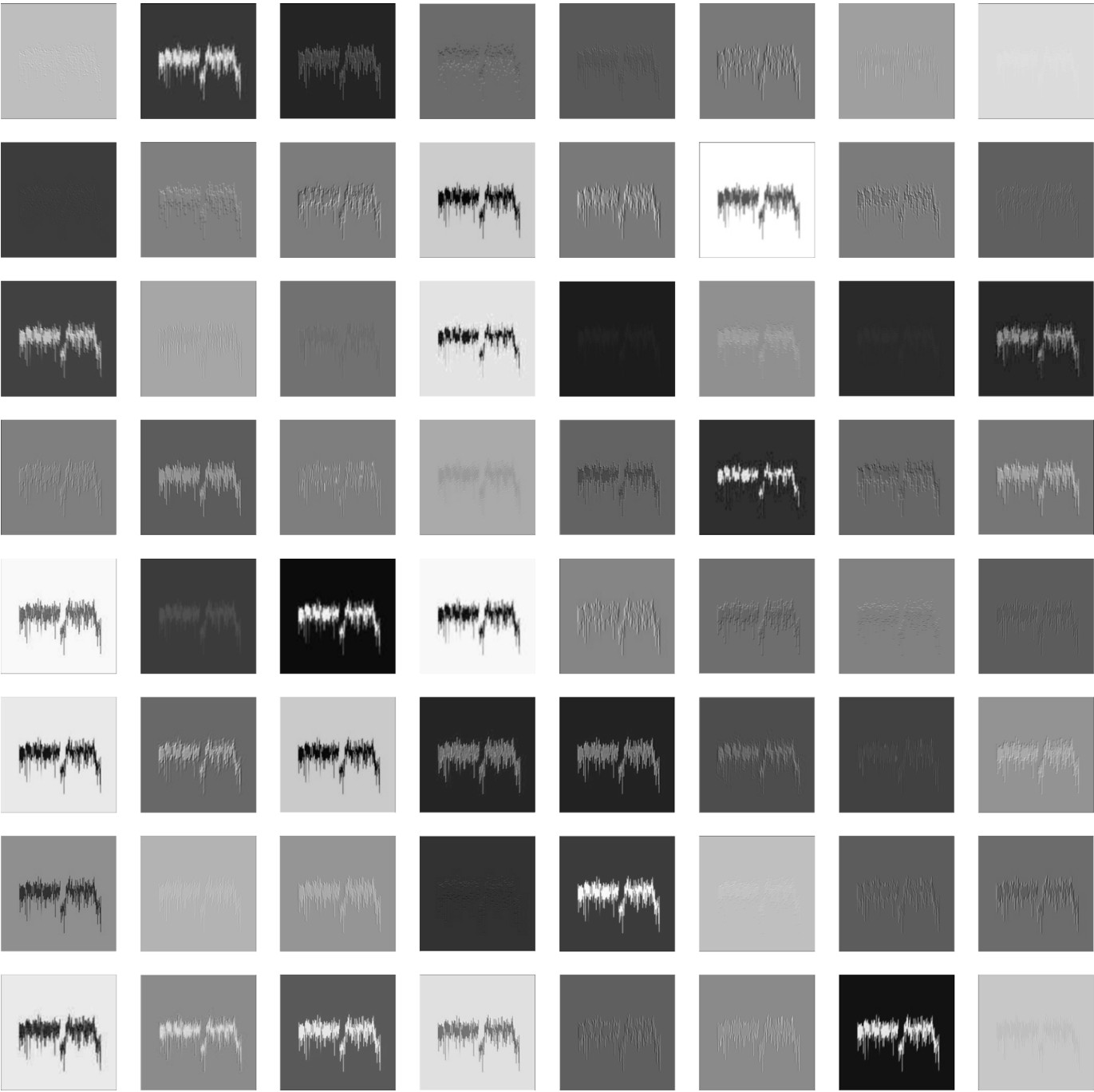

**Figure 15.** Feature Map Extraction Convolutional Layer 0.

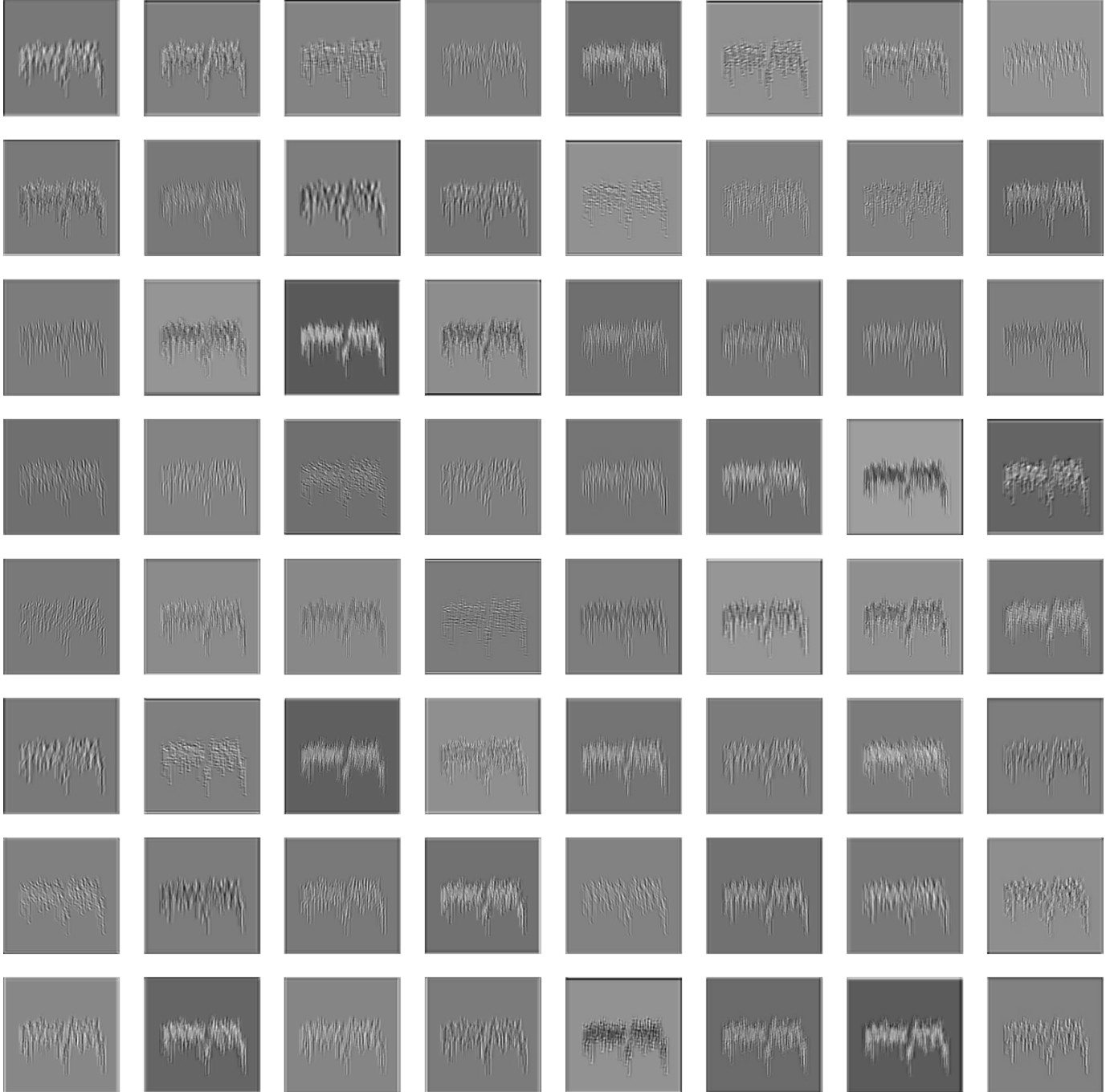

**Figure 16.** Feature Map Extraction Convolutional Layer 20.

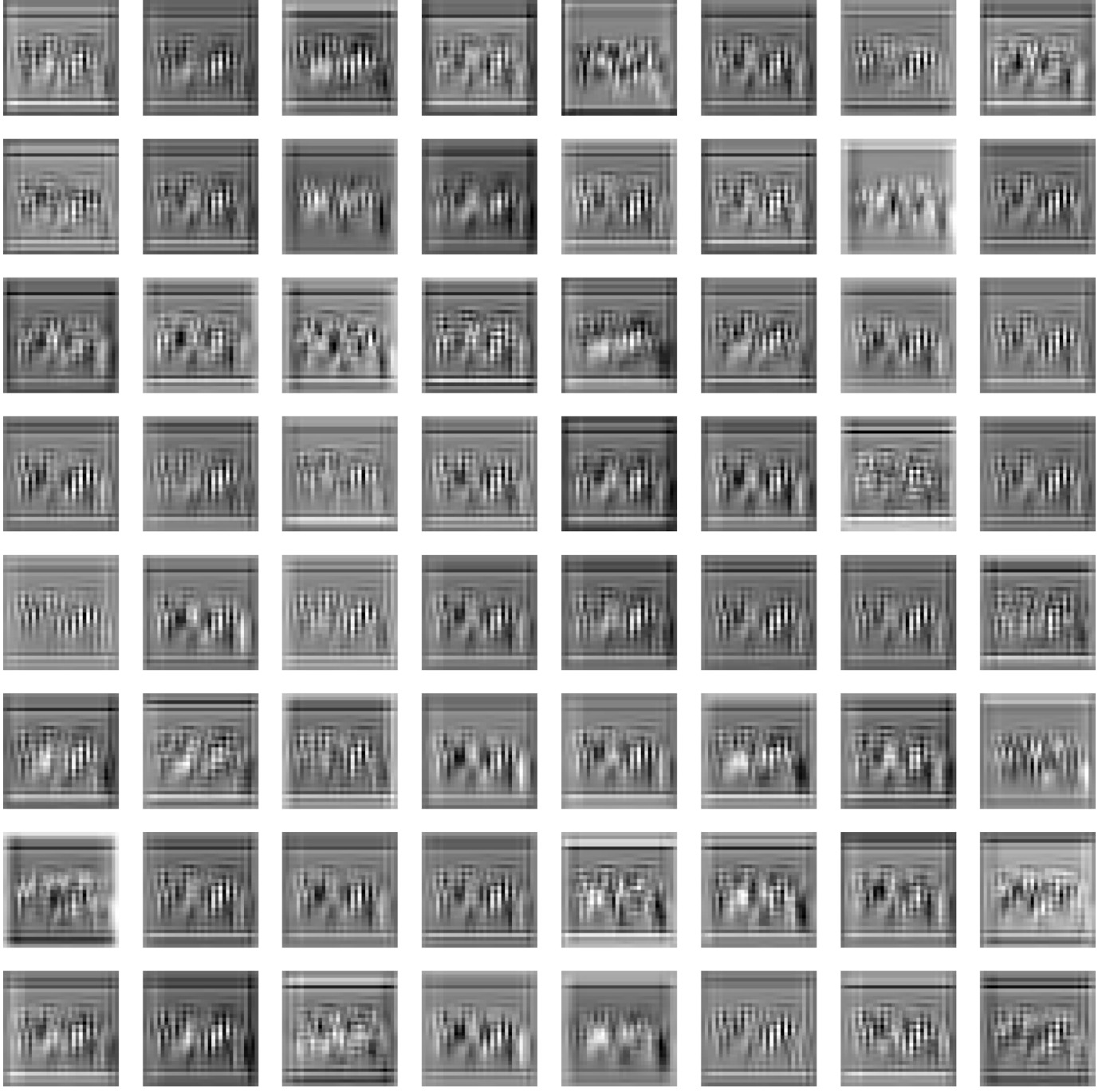

**Figure 17.** Feature Map Extraction Convolutional Layer 48.

*3.2. Classifier Results*

3.2.1. Cross Validation Training/Test Data

It is clear from Table 2 that PSD outperforms raw constellation, spectrogram and histogram representations.

**Table 2.** UAV Classification Accuracy (%) and F1-Score (%).

| Metric | Raw | Spec | PSD | Hist |
|---|---|---|---|---|
| Acc | 45.3 (+/−1.1) | 83.8 (+/−1.1) | 92.3 (+/−0.3) | 37.0 (+/−0.2) |
| F1 | 45.1 (+/−1.1) | 83.7 (+/−1.2) | 92.3 (+/−0.3) | 36.8 (+/−0.2) |

Spectrogram representation was approximately 10% less accurate than PSD, with histogram performing the worst out of all the representations. PSD produced 92.3 (+/−0.3%) accuracy and F1-score which is an increase of over 45% from previous published work.

Table 3 shows the individual representations and their F1-score performance for each individual class. PSD outperforms the other representations in all classes except for Bebop Mode 4 (flying with video). Spectrogram was 4% more accurate at classifying Bebop Mode 4. Overall Table 3 shows that PSD is the most accurate way to classify UAV signals across 80 MHz of the Wi-Fi band using transfer learning with ResNet50 CNN feature extraction and LR.

**Table 3.** Individual Classification LR F1-Score (%).

| Mode | Raw | Spec | PSD | Hist |
|---|---|---|---|---|
| No UAV | 51 | 97 | 100 | 49 |
| Bebop Mode 1 | 26 | 88 | 97 | 26 |
| Bebop Mode 2 | 29 | 83 | 97 | 18 |
| Bebop Mode 3 | 90 | 79 | 100 | 79 |
| Bebop Mode 4 | 23 | 87 | 83 | 17 |
| AR Mode 1 | 23 | 92 | 100 | 21 |
| AR Mode 2 | 31 | 86 | 94 | 14 |
| AR Mode 3 | 20 | 69 | 71 | 18 |
| AR Mode 4 | 99 | 72 | 100 | 100 |
| Phantom Mode 1 | 35 | 64 | 71 | 25 |

### 3.2.2. Hold-Out Evaluation Results

The evaluation data set results in terms of accuracy and F1-score can be seen in Table 4. These results confirm the cross validation results in Table IV with PSD producing the highest accuracy and F1-score. Table 4 shows that PSD is over 10% more effective than spectrograms and over 40% more accurate than raw constellation and histograms.

**Table 4.** Evaluation Data Accuracy (%) F1-Score (%).

| Metric | Raw | Spec | PSD | Hist |
|---|---|---|---|---|
| Acc | 43.1 | 81.5 | 91.2 | 36.7 |
| F1 | 42.9 | 81.7 | 91.2 | 36.6 |

Figures 13 and 14 show the confusion matrix for PSD and spectrogram representations, respectively. Both are able to detect whether a UAV is present or not with an accuracy of 96% or over, with PSD performing at 99.7%.

Figures 18 and 19 confirm that both representations were worst at classifying the AR in mode 3 (flying without video) and the Phantom 3 when switched on and connected to the controller. The reason for the Phantom 3 could be the fact that it will hop Wi-Fi channels based on interference. Without monitoring the spectrum separately we can't be sure whether this is happening. The AR when flying without video must look similar in terms of features in the frequency domain to the Phantom 3 when switched on and connected to the controller.

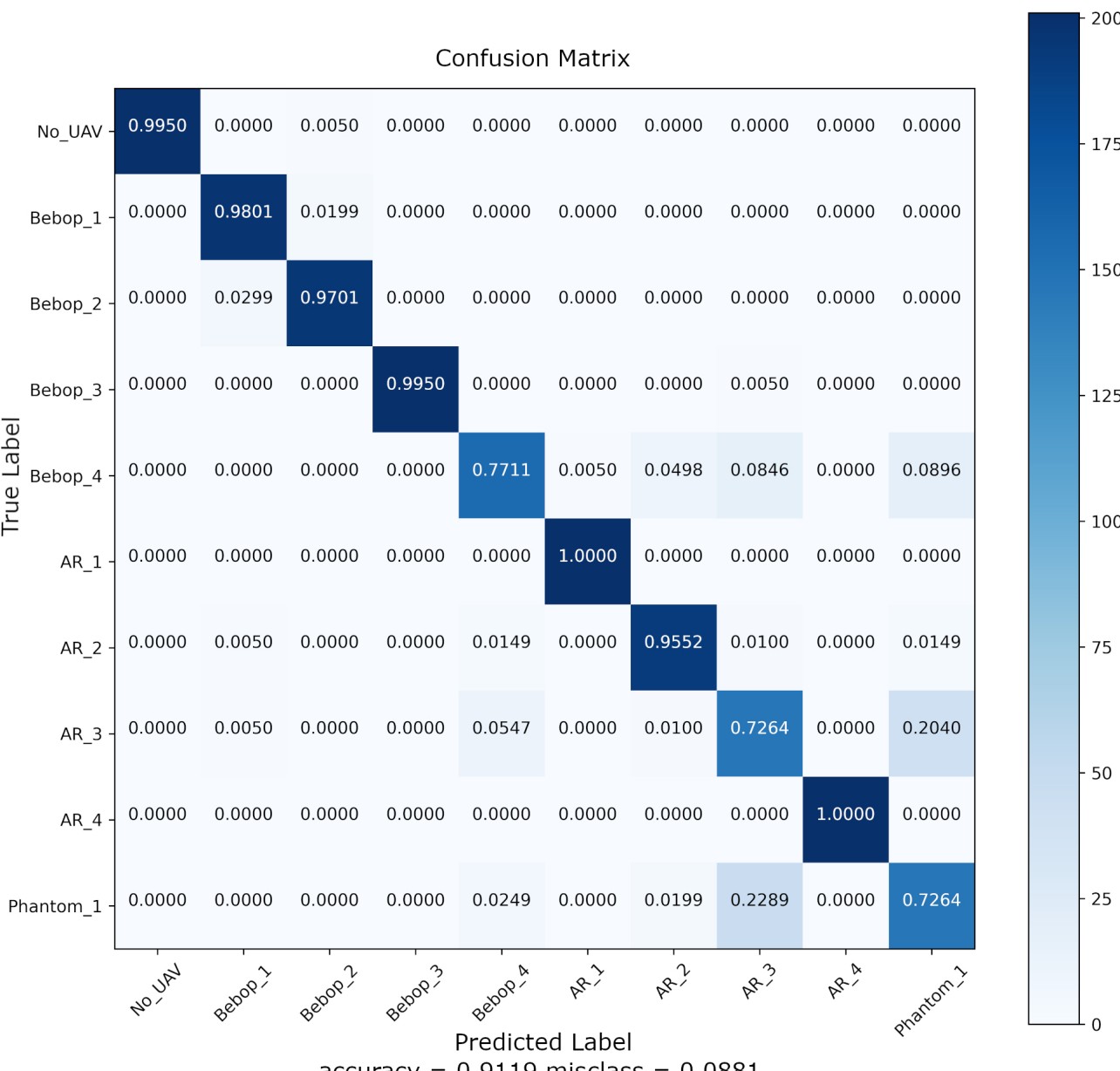

**Figure 18.** Confusion Matrix PSD.

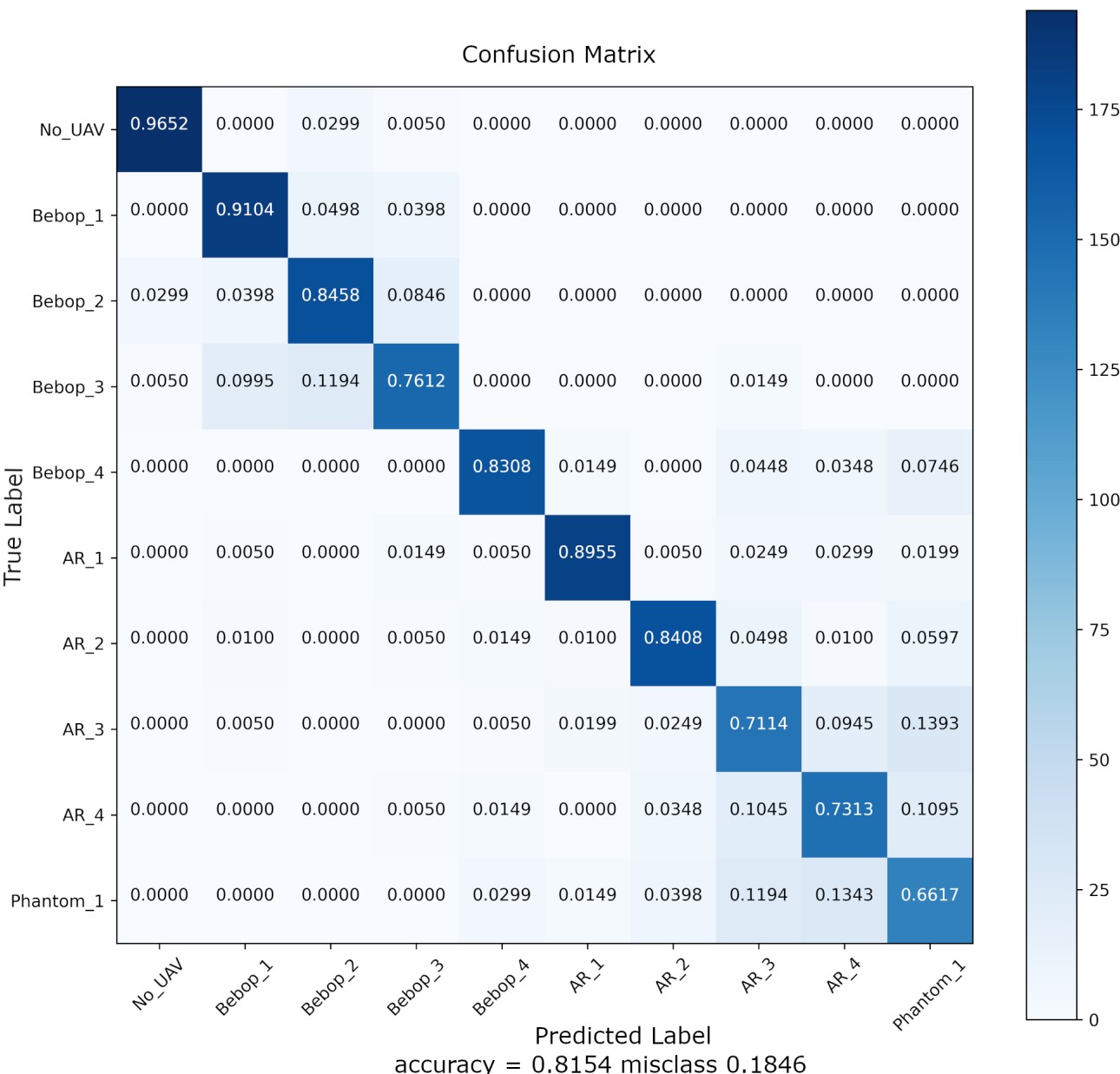

**Figure 19.** Confusion Matrix Spectrogram.

Figures 18 and 19 show the confusion matrix for PSD and Spectrogram representations. Both representations produce an overall accuracy of above 81% but PSD performs with higher overall accuracy above 91%. Due to the fact we produced our results with the same open DroneRF dataset used and produced by Al-Sa'd et al. [22], who achieved accuracy of 46.8% using a DNN across all 10 classes, we can directly compare our results. Our results show that LR with PSD, to achieve over 91% accuracy, a 45% increase compared with the prior work. We achieve this by viewing UAV classification as an image classification problem and utilising transfer learning from the field of imagery. Al-S'ad et al. found that when they increased the classification from detecting the presence of a UAV (2 class) to its type (4 class), to include flight modes (10 class) the accuracy decreased significantly. They put this down to similarities caused by two of the UAVs (Bebop and AR) being manufactured by the same company. We have shown our approach using CNN feature extraction able to improve these results distinguishing between same manufacturer.

## 4. Conclusions

Our results have shown that PSD outperforms raw constellation, spectrogram and histogram representations for LR. PSD produced over 91% accuracy with cross validation results and the evaluation dataset. We achieve this by viewing UAV classification as an image classification problem, utilising transfer learning and presenting signal representations as graphical images to a deep CNN. If a system like this was employed in the real world it would likely need to be trained in the particular environment that it needed to work in. For example, a built up city area will have more background noise in the Wi-Fi bands than a rural area. This will also likely affect the accuracy of the classifier so field testing in city areas is paramount for this type of system. It may also help researchers understand how much frequency hopping occurs due to interference and whether this has an impact on detection and classification accuracy. Further, the issue of how often you would need to re-train the classifier, as RF bands are constantly changing, and how much change the classifier can cope with before accuracy starts being affected is an important question which would need further investigation.

Future work could also consider the employment of another SDR to capture the 5 Ghz band to fully represent dual frequency band UAVs such as the Bebop and Phantom 3. It is thought that this would further improve accuracy as it provides an increase of distinguishing features. The dataset could also be expanded to include more UAV platforms. This method which utilises signal representations as graphical images would require more processing power and therefore increased energy requirements compared with processing 1D data. Further work could look at hardware implementations such as FPGA, GPU and hardware accelerators such as Tensor processing unit [45] by Google to evaluate practical limitations for 2D data against the use of 1D compared with accuracy of the models. In conclusion our results have shown a novel approach by treating UAV classification as an imagery detection problem utilising the benefits of transfer learning and outperforming previous work in the field by over 45%.

**Author Contributions:** Conceptualization, C.J.S. and J.C.W.; methodology, C.J.S.; software, C.J.S.; validation, C.J.S. and J.C.W.; investigation, C.J.S.; resources, C.J.S. and J.C.W.; data curation, C.J.S.; writing—original draft preparation, C.J.S.; writing—review and editing, C.J.S. and J.C.W.; visualization, C.J.S.; supervision, J.C.W.; project administration, C.J.S. and J.C.W. All authors have read and agreed to the published version of the manuscript.

**Funding:** This research received no external funding.

**Data Availability Statement:** Data not yet publicly available.

**Acknowledgments:** This work was carried out through the support of the School of Computer Science and Electronic Engineering, University of Essex, UK and the Royal Air Force, UK.

**Conflicts of Interest:** The authors declare no conflict of interest.

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
