# Peer review of "Unmanned Aerial Vehicle Operating Mode Classification Using Deep Residual Learning Feature Extraction"

_aerospace, doi:10.3390/aerospace8030079_

Round 1

Reviewer 1 Report

This paper has introduced a novel approach which considers UAV detection as an imagery classification problem. Authors have utilised imagery transfer learning via a ResNet50 deep Convolution Neural Network (CNN) for feature extraction. They have considered various signal representations including Power Spectral Density (PSD); Spectrogram, Histogram and raw IQ constellation. Authors have compared their performance through machine learning classifier Logistic Regression (LR). Three popular UAVs were classified in different modes; switched on; hovering; flying; flying with video; and no UAV present, creating a total of 10 classes. Results obtained by the authors have been validated with 5-fold cross validation and an independent dataset, show PSD with LR to produce over 91% accuracy for 10 classifications. This paper treats UAV detection as an imagery classification problem, utilising the benefits of transfer learning through CNN feature extraction and outperforming previous work in the field. I rate the article very highly. All models and calculations do not raise any objections. The literature is right. The individual chapters form a logical whole. The presented results are of great practical and theoretical importance. The work can be printed in the form presented by the authors.

Author Response

Many thanks for your comments and support.

Kindest regards,

Carolyn

Reviewer 2 Report

This paper presents a method of drone detection by representing RF Drone detection data as images and using CNN-based transfer learning for feature extraction on these images. The idea of representing signals as images and then using Computer Vision based methods on these images is interesting, but it needs a bit more justification in the paper.

The following issues must be addressed:

  1. The author’s previous work in the area is not referenced here [1] and there is significant overlap. This looks like just a timing issue but should be updated now, clarifying the added contribution here.
  2. The abstract should clearly state what this design does. The term “imagery” is used a lot, but it is not clear what this means. Is it a photograph of a drone? I think not.
  3. The introduction needs tightening up. It is good at giving context, but by the time we get to what the paper is about (line 119), it is still not clear.
  4. Why CNNs? These are effectively 1D signals. Deep learning methods can be used on these signals without treating them as images. Doing so will lead to less efficiency and greater computational complexity and so greater processing requirements so why do it?
  5. This image representation of signals has been done before in other contexts, e.g. [2], but these are not referenced. Context is needed to clarify where the novelty lies.
  6. Efficiency would be a major issue which would affect the practicality of implementing this, e.g. in hardware. This should at least be addressed.
  7. Ditto computational complexity and processing power. Deep learning solutions, particularly image-based, can be heavy on both processing and energy requirements. Some context on where/how this would be implemented might help address these two points.
  8. What is the resolution of the images? Does one pixel equal one data point? It should.

The following are minor issues but must be fixed:

  1. Citation style is a little inconsistent and some citations are incomplete. Check this.
  2. David Cameron was not Prime Minister of the UK in 2017. These news references are incomplete and do not include the publication.
  3. Line 281 runs off the page.

If these issues are addressed, this paper would be suitable for publication in Aerospace.

[1]   C. J. Swinney and J. C. Woods, “Unmanned Aerial Vehicle Flight Mode Classification using Convolutional Neural Network and Transfer Learning,” in 2020 16th International Computer Engineering Conference (ICENCO), Dec. 2020, pp. 83–87, doi: 10.1109/ICENCO49778.2020.9357368.

[2]   H. Long, L. Sang, Z. Wu, and W. Gu, “Image-Based Abnormal Data Detection and Cleaning Algorithm via Wind Power Curve,” IEEE Trans. Sustain. Energy, vol. 11, no. 2, pp. 938–946, Apr. 2020, doi: 10.1109/TSTE.2019.2914089.

Author Response

Many thanks for your time and the valuable feedback you have given us. Please see below for details of how we have incorporated the revisions. 

The author’s previous work in the area is not referenced here [1] and there is significant overlap. This looks like just a timing issue but should be updated now, clarifying the added contribution here.

Thank you for highlighting this. Yes it was a timing issue, the conference paper was published not long after we submitted this. We have added in the following:

Lines 105 – 112

“Swinney and Woods [23] use a VGG-16 CNN for feature extraction after using the DroneRF dataset to produce Power Spectral Density and Spectrogram signal representations. They evaluate machine learning classifiers Support Vector Machine, Logistic Regression and Random Forest and achieve 100% accuracy for 2 class detection, 88.6% for 4 class UAV type classification and 87.3% accuracy for 10 flight mode classifications. Swinney and Woods show treating signal representations as images using a CNN that they are able to distinguish between UAVs produced by the same manufacturer.”

Lines 124-126

“Our work in this paper will consider various signal representations and will also carry on the findings of Swinney and Woods [23] by investigating a deeper CNN architecture.”

The abstract should clearly state what this design does. The term “imagery” is used a lot, but it is not clear what this means. Is it a photograph of a drone? I think not.

Thank you for highlighting this point. Updated to emphasis that the CNN is receiving signal representations as images and that transfer learning is used to potentially mitigate the need for a large signal dataset. Re-worded lines 5-7 as follows:

“We consider signal representations Power Spectral Density (PSD); Spectrogram, Histogram and raw IQ constellation as graphical images presented to a deep Convolution Neural Network (CNN) ResNet50 for feature extraction. Pre-trained on ImageNet, transfer learning is utilised to mitigate the requirement for a large signal dataset.”

Further taking on the point about clarifying the term “imagery”, we have also changed the following:

Lines 131-137

“The approach proposed in this work utilises transfer learning through the use of a pre-trained ResNet50 CNN on ImageNet to extract features from our graphical image datasets of the various signal representations. These features are then classified using machine learning classifier LR. Figure \ref{fig:1} shows the process from the raw signal data in the open DroneRF dataset in block 1 through to the classification of the signal. Block 2 refers to plotting the raw data as Spectrogram, Histogram, Raw IQ constellation and Power Spectral Density (PSD) graphical representations.”

Lines 368-392

“Our results have shown that PSD outperforms raw constellation, spectrogram and histogram representations for LR. PSD produced over 91\% accuracy with cross validation results and the evaluation dataset. We achieve this by viewing UAV classification as an image classification problem, utilising transfer learning and presenting signal representations as graphical images to a deep CNN.”

The introduction needs tightening up. It is good at giving context, but by the time we get to what the paper is about (line 119), it is still not clear.

Re-scrubbed introduction, altered into the following structure -

  • UAVs being a security issue
  • Detection and mitigation needed to address issue & methods to do this
  • Commercial systems
  • Current literature
  • Literature this work leads directly on from
  • Introduction to approach

Specific changes include –

Removed –

“with energy being the largest industry and good delivery the fastest growing [3]. It is predicted that over 76,000 UAVs for government and commercial use will be operational by 2030 but there are no accurate estimates of privately owned UAVs in the UK [4]”

Altered from –

Earlier that year 12 offenders were sentenced for targeting prisons in Birmingham, Wolverhampton, Worcestershire, Warrington, Lancashire and Liverpool for 55 drug deliveries worth over £500,000 and this was done using UAVs [7].

To – Line 28 -

“UK prisons have been targeted for drug deliveries using UAVs [5].”

Removed –

“The ease of adaptation can be seen in many industries. In 2020 a Chinese volunteer group successfully equipped a UAV with a petrol tank to produce a flamethrower able to eradicate wasps nests [8].”

Removed –

“General Raymond, Commander of US Special Operations Command in May 2017 was quoted to say, "About five or six months ago, there was a day when the Iraqi effort nearly came to a screeching halt, where literally over 24 hours there were 70 drones in the air" [11]. He was referring to quadcopters, easily purchased at low cost, which had been weaponised to fire 40mm munitions.”

Removed –

“In October 2019 the Commons Science and Technology Committee recommended the malicious use of UAVs to be an intelligence priority for the Ministry of Defence [13].”

Added – Line 67-69 -

“This paper looks to extend current research in the use of RF signals to detect and classify UAVs. We will now discuss current and associated literature in the field and systems available commercially.”

Moved commercial systems to come before current literature (lines 70-77).

Re-worded –

“Al-S’ad et al. [24] use USRP SDR to detect 3 different UAVs operating in different modes - switched on; hovering; flying with video; and flying without video. The signal is then processed using discrete Fourier transform and results are input into a deep neural network (DNN). Results show the classification accuracy to drop significantly when the classes are increased; 99.7% for 2 classes (UAV present or not), 84.5% for 4 classes (UAV type) and 46.8% for 10 classes (UAV type and flight mode). The DroneRF [25] is the first open dataset for the classification of UAVs so has made a significant contribution to the field.”

To – lines 96-105 –

“Al-S’ad et al. produce the open DroneRF [21] dataset. A significant contribution to the field, this is the first open dataset for the classification of UAV flight modes. In the associated publication, Al-S’ad et al. [22] use the USRP SDR to capture raw IQ data and a deep neural network (DNN) to classify 3 different UAVs operating in different modes - switched on; hovering; flying with video; and flying without video. Results show classification accuracy to drop significantly when the classes are increased; 99.7% for 2 classes (UAV present or not), 84.5% for 4 classes (UAV type) and 46.8% for 10 classes (UAV type and flight mode). Al-S’ad et al. [22] struggle to classify flight modes accurately using raw data and a DNN, they conclude that an issue may exist with distinguishing between UAVs produced by the same manufacturer.”

This image representation of signals has been done before in other contexts, e.g. [2], but these are not referenced. Context is needed to clarify where the novelty lies.

Thank you for high-lighting this. We have added in several references to the classification of wireless signals which are also using the same process of images and CNNs – however, only considering spectrogram representations as 2D images.

Added lines 113-131

“Other domains have benefitted from considering a signal as an image. Long et al. [24] use images of wind power curves to detect anomalies in wind turbine data by identifying anomalous data points. They show the method superior to more traditional methods of outlier detection such as k-means. Spectrogram signal representations as images are used for jamming detection in [25] by doing a comparison with a baseline image. This work does not extend to classification of the signal. O’Shea et al. [26] also use spectrograms as images in conjunction with CNN feature extraction to classify wireless signals such as GSM and Bluetooth. Their work concludes that this method struggles to pick up burst communications as the spectrogram is looking at the frequency changes over a set time period. If the burst didn’t happen within the time frame the spectrogram would miss it. In a similar vein UAV signals can hop around the frequency spectrum, potentially making signals harder to detect and classify accurately through spectrogram time domain image representation. Due to the significance of this method in wireless communications, our work in this paper will extend the types of signal representations considered as graphical images presented to a CNN. Viewing a signal in 2D as an image over 1D signal data allows a human in the loop to visually identify issues, in some cases providing a contextual understanding. We will also carry on the findings of Swinney and Woods [26], who showed flight mode classification possible with the same manufacturer, by investigating a deeper CNN architecture for classification accuracy.”

Why CNNs? These are effectively 1D signals. Deep learning methods can be used on these signals without treating them as images. Doing so will lead to less efficiency and greater computational complexity and so greater processing requirements so why do it?

Thank you for bringing this point to our attention. It has been investigated successfully particularly in wireless communications with 2D spectrograms being presented as images to a CNN. The main advantage is visualisation - allowing a human to provide some contextual understanding to what is going on. Taking your comments on board we have added the following revisions -

Added in lines 125-131

“Due to the significance of this method in wireless communications, our work in this paper will extend the types of signal representations considered as graphical images presented to a CNN. Viewing a signal in 2D as an image over 1D signal data allows a human in the loop to visually identify issues, in some cases providing a contextual understanding.”

Efficiency would be a major issue which would affect the practicality of implementing this, e.g. in hardware. This should at least be addressed. Ditto computational complexity and processing power. Deep learning solutions, particularly image-based, can be heavy on both processing and energy requirements. Some context on where/how this would be implemented might help address these two points.

Really significant point, thank you for raising it and for your insight into the issue. We have added the following lines to address that the important issue of efficiency and processing power is not within the scope of this study. However, we have referenced a piece of work which considers specifically hardware issues with deep CNNs and future technology. We have also included the insight as a consideration for further work in the discussion.

Line 132-126 -

“This work does not consider or compare the additional processing power that would be required for the use of a deep CNN compared to the use of 1D data. For example, there would be practical limitations on hardware. However, this is a larger issue with the implementation of DNNs for practical real time and embedded applications which is reviewed in [27].”

Lines 412 - 417

“This method, which utilises signal representations as graphical images would require more processing power and therefore increased energy requirements compared with processing 1D data. Further work could look at hardware implementations such as FPGA, GPU and hardware accelerators such as Tensor processing unit [45] by Google to evaluate practical limitations for 2D data against the use of 1D compared with accuracy of the models.”

What is the resolution of the images? Does one pixel equal one data point? It should.

Thank you for your insight into this. The dataset is at a higher resolution which does represent all the points and the sub sampling operations maintain the essence of the graph but without the computational overhead. Taking on board your comments we have adjusted the following text –

Lines 262-268

“The DroneRF dataset includes 10.25s of recording with no UAV present and 5.25s for each UAV flight mode at a sample rate 200MS/s, producing a dataset larger than 40GB. Samples are plotted in each of the 4 signal representations using MatPlotlib and saved as images with 300 DPI. We constructed separate datasets of images for raw constellation, spectrogram, PSD and histogram. Each class within each dataset contained 1000 image representations resized to 224x224 pixels. The databases were split so there were 8,000 images for use with k-fold cross validation and 2000 images with the evaluation set.”

Citation style is a little inconsistent and some citations are incomplete. Check this.

Have altered all to same style in the bib file.

David Cameron was not Prime Minister of the UK in 2017. These news references are incomplete and do not include the publication.

Thank you for highlighting this - re-worded to former UK Prime Minister. Lines 31-33.

“As far back as 2017 former UK Prime Minister David Cameron, warned of the risk of the use of a UAV equipped with an aerosol device for the dispersal of nuclear or chemical material over a European city [7].”

Line 281 runs off the page.

This was due to the length of Limited memory Broyden–Fletcher–Goldfarb–Shanno (LGBFS) and this has now been adjusted for. Please see line 302.